# Deep learning the collisional cross sections of the peptide universe from a million experimental values

Florian Meier [1,5,6], Niklas D. Köhler[2,6], Andreas-David Brunner [1,6], Jean-Marc H. Wanka[2], Eugenia Voytik[1], Maximilian T. Strauss [1], Fabian J. Theis [2,3✉] & Matthias Mann [1,4✉]

The size and shape of peptide ions in the gas phase are an under-explored dimension for mass spectrometry-based proteomics. To investigate the nature and utility of the peptide collisional cross section (CCS) space, we measure more than a million data points from whole-proteome digests of five organisms with trapped ion mobility spectrometry (TIMS) and parallel accumulation-serial fragmentation (PASEF). The scale and precision (CV < 1%) of our data is sufficient to train a deep recurrent neural network that accurately predicts CCS values solely based on the peptide sequence. Cross section predictions for the synthetic ProteomeTools peptides validate the model within a 1.4% median relative error (R > 0.99). Hydrophobicity, proportion of prolines and position of histidines are main determinants of the cross sections in addition to sequence-specific interactions. CCS values can now be predicted for any peptide and organism, forming a basis for advanced proteomics workflows that make full use of the additional information.

[1] Department Proteomics and Signal Transduction, Max Planck Institute of Biochemistry, Martinsried, Germany. [2] Institute of Computational Biology, Helmholtz Zentrum München—German Research Center for Environmental Health, Neuherberg, Germany. [3] Department of Mathematics, TU München, Munich, Germany. [4] NNF Center for Protein Research, Faculty of Health Sciences, University of Copenhagen, Copenhagen, Denmark. [5] Present address: Functional Proteomics, Jena University Hospital, Jena, Germany. [6] These authors contributed equally: Florian Meier, Niklas D. Köhler, Andreas-David Brunner. ✉email: fabian.theis@helmholtz-muenchen.de; mman@biochem.mpg.de

The combination of ion mobility spectrometry (IMS) and mass spectrometry (MS) extends conventional liquid chromatography–mass spectrometry (LC–MS) by an extra dimension of separation, increasing peak capacity, selectivity, and depth of analysis[1–5]. Recent advances have greatly improved the sensitivity of commercially available IMS devices and the technology is now set for a broader application in MS-based proteomics[6–10].

IMS separates ions in the gas phase (typically in the mbar pressure range) based on their size and shape within milliseconds. This time scale allows recording full ion mobility spectra between typical chromatographic peaks (seconds) and the acquisition pulses of time-of-flight (TOF) instruments (~100 µs). We have recently integrated trapped ion mobility spectrometry (TIMS)[11,12], a relatively new and particularly compact ion mobility device, with a high-resolution quadrupole TOF mass analyzer[10,13,14]. In MS/MS mode, this opens up the possibility to step the precursor selection window as a function of ion mobility, allowing the fragmentation of multiple precursors during a single TIMS scan[13]. We termed this novel scan mode parallel accumulation-serial fragmentation (PASEF) and demonstrated that it increases MS/MS rates more than ten-fold without any loss in sensitivity as is otherwise inherent to faster scanning rates[10,15].

An intriguing feature of the combination of TIMS and PASEF is that it should allow the acquisition of ion mobility values on a very large scale. Such data have previously been measured on a case by case basis by classical drift tube IMS, in which a weak electric field drags ions through an inert buffer gas[16–18]. Larger ions collide more frequently with gas molecules and hence traverse the drift tube with a lower speed as compared with their smaller counterparts. In TIMS the physical process is the same, except that the setup is reversed with the electric field holding ions stationary against an incoming gas flow, prior to their controlled release from the device by lowering the electric field[19,20]. In both cases, the measured ion mobility (reported as the reduced ion mobility coefficient $K_0$) can be used to derive a collisional cross section (CCS), which is the rotational average of an ion's gas-phase conformation[21,22]. The CCS intrinsically depends on the ion structure, which is also illustrated by the fact that different classes of biomolecules (e.g., metabolites, carbohydrates, peptides) show different trends in their ion mobilities as a function of molecular mass[23]. Interestingly, conformations also vary within a compound class - to the extent that isobaric peptide sequences can be distinguishable by their different CCS[24,25].

The link between the amino acids of a peptide and its measured cross section has the potential to increase the confidence in its identifications through reference or predicted CCS values. This has motivated researchers to develop various (machine learning) models based on amino acid-specific parameterization and physicochemical properties[16,26–29]. However, as comprehensive experimental data are not available, predicting the full complexity of the peptide conformational space remains elusive. Furthermore, it is not clear which properties should be considered to best parameterize such models and make them generalizable. We reasoned that a combination of very large and consistent datasets acquired by PASEF with state of the art deep learning methods would address both challenges. Due to their inherent flexibility and their ability to scale to large datasets, deep learning methods have proven very successful in genomics[30,31] and more recently in proteomics for the prediction of retention times and fragmentation spectra[32–35].

We here set out to explore the nature and utility of the peptide CCS space in proteomics by first measuring a very large dataset of CCSs by TIMS-TOF PASEF across five different biological species. Building on this dataset, we develop and train a bidirectional recurrent neural network with long short-term memory (LSTM) units to predict CCS values for any peptide sequence in the tryptic peptide universe. Interpreting our network based on recent approaches from explainable AI allows us to investigate the nature of the underlying relationship between linear peptide sequence and peptide cross section.

## Results

**Construction of a very large-scale peptide CCS dataset**. To fully capture the conformational diversity of peptides in the gas phase, we generated peptides from whole-cell proteomes of *Caenorhabditis elegans*, *Drosophila melanogaster*, *Escherichia coli*, *HeLa*, and *budding yeast* using up to three different enzymes with complementary cleavage specificity (trypsin, LysC, and LysN). To increase the depth of our analysis, we split peptide mixtures into 24 fractions per organism and analyzed each of them separately with PASEF on a TIMS-quadrupole TOF MS (Methods; Fig. 1a). As this is the same setup we used before, we combined our new experimental data with our previously reported dataset from a tryptic HeLa digest[10].

In total, we compiled 360 LC-MS/MS runs and processed them in the MaxQuant software[36,37]. This resulted in about 2.5 million peptide spectrum matches and 426,845 unique peptide sequences at globally controlled false discovery (FDR) rates of less than 1% at the peptide and protein levels for each organism and enzyme. MaxQuant links each peptide spectrum match to a four-dimensional (4D) isotope cluster (or 'feature') in mass, retention time, ion mobility, and intensity dimension. For each of these, the ion mobility value is determined as the intensity-weighted average of the corresponding mobilogram trace and can be converted into an ion-neutral CCS value using the Mason-Schamp equation[21]. Some peptides occur in more than one conformation and have multiple peaks in an LC-TIMS-MS experiment, but for simplicity we here chose to keep only the most abundant feature per charge state (Supplementary Fig. 1).

Overall, our dataset comprises over two million CCS values, which we collapsed to about 570,000 unique combinations of peptide sequence, charge state and, if applicable, side chain modifications such as oxidation of methionine (Fig. 1b). Peptide sequence lengths ranged from 7 up to 55 amino acids with a median length of 14. The trypsin and LysC datasets contributed 79% of the peptide sequences (C-terminal R or K), whereas LysN peptide (N-terminal K) accounted for the remaining 21%. Within the two classes of peptides, the proportion of the terminal amino acids conformed to their expected frequencies from the database (Fig. 1c, d). Due to our selection of enzymes, peptides should have at least one basic amino acid. Consequently, singly charged ions were a small minority (2%), which we excluded from further analysis. We detected 69% of the peptides in the doubly charged, and 25% in the triply charged and 4% in the quadruply charged state. Plotting the mass-to-charge (*m/z*) vs. CCS distribution of all peptides separates them by their charge state over the *m/z* range 400–1700 Å$^2$ and 300–1000 Å$^2$ in cross section (Fig. 1e). Within each charge state, *m/z* and CCS were correlated in accordance with previous observations in smaller datasets[10,18,23,38–40]. Overall, 95% of all tryptic peptides were distributed within ±8% around power-law trend lines for each charge state (Supplementary Fig. 2). Interestingly, the deviation increases with charge state and mass—to the extent that there are two distinct sub-populations for charge state 3—perhaps due to the increased amino acid variability and structural flexibility in longer sequences. Our data show that peptides occupy about one-quarter of the 2D *m/z*-mobility space, whereas a fully orthogonal 2D separation would occupy the full space. Assuming an average ion mobility resolution of 60, this translates into an at least ten-fold increased analytical peak capacity as compared with only MS (Supplementary Fig 3).

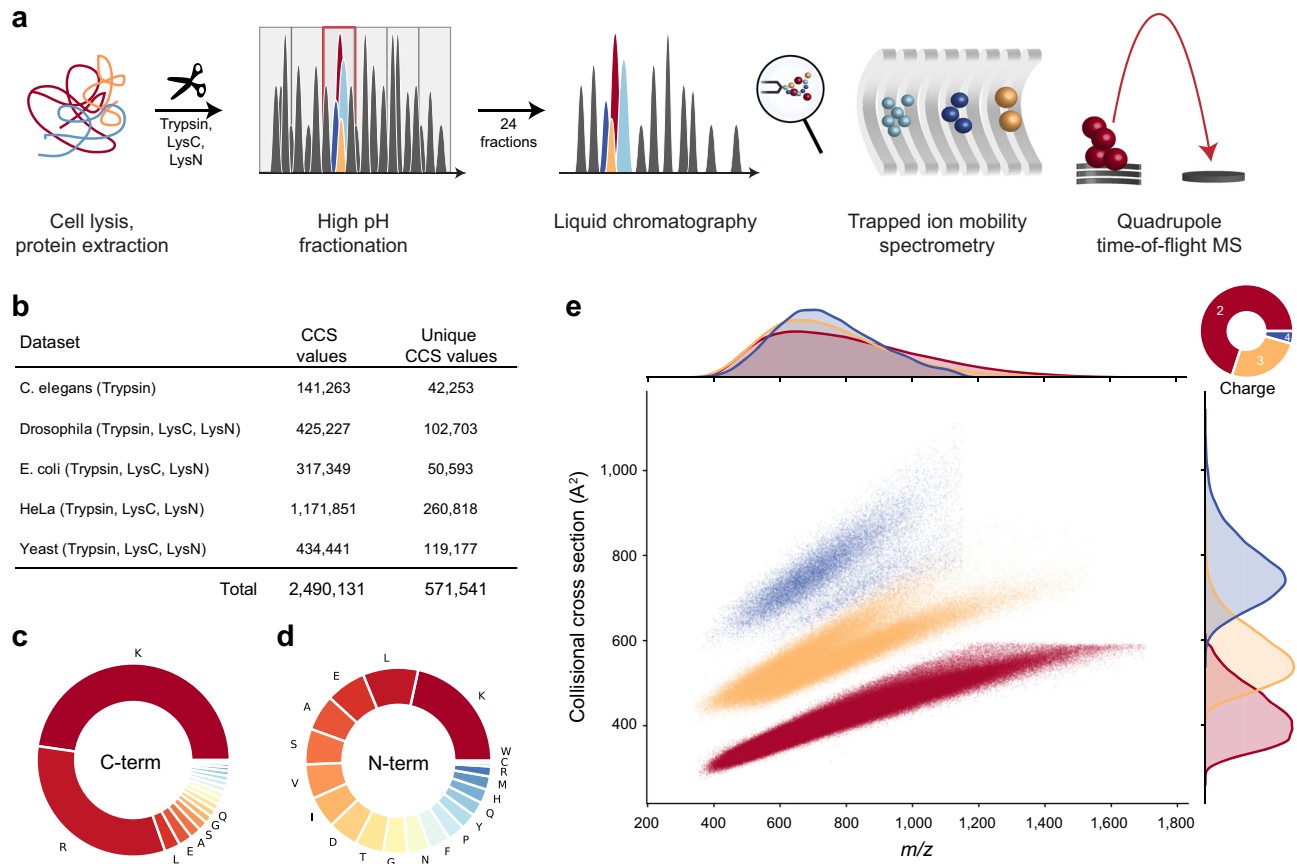

**Fig. 1 Large-scale peptide collisional cross section (CCS) measurement with TIMS and PASEF. a** Workflow from extraction of whole-cell proteomes through digestion, fractionation, and chromatographic separation of each fraction. The TIMS-quadrupole TOF mass spectrometer was operated in PASEF mode. **b** Overview of the CCS dataset in this study by organism. **c** Frequency of peptide C-terminal amino acids. **d** Frequency of peptide N-terminal amino acids. **e** Distribution of 559,979 unique data points, including modified sequence and charge state, in the CCS vs. *m/z* space color-coded by charge state. Density distributions for *m/z* and CCS are projected on the top and right axes, respectively. Source data are provided as a Source Data file.

**Evaluating the accuracy, precision, and utility of TIMS CCS measurements**. Peak capacity indicates how many peptides can be analytically resolved from each other. However, for their identification it is sufficient to determine their apex positions with adequate precision. In MS-based proteomics, accurate measurement of the peptide mass greatly reduces the number of candidates in database searches[36], and the retention time can likewise be employed as a filter, as is typically done in the analysis of data-independent acquisition (DIA) experiments[41]. We reasoned that ion mobility values should be precise and reproducible as they are based on gas-phase interactions and defined electric fields, in contrast to chromatographic retention times, which depend on surface interactions that vary according to sample matrices and over time. We therefore investigated the precision, accuracy and added benefit of ion mobility measurements in our dataset.

First, we calculated correlation coefficients for retention times and CCS values from pair-wise overlapping tryptic peptides in the 168 LC-MS/MS runs that had the highest number of shared peptides across organisms. Depending on evolutionary distance, this number ranged from none to hundreds and these formed the basis of our calculations. We obtained two triangular half-matrices of color-coded Pearson correlation coefficients—one for the retention time correlations and one for CCS (upper and lower part of Fig. 2a, respectively). Correlation values were generally above 0.9 for both retention time and cross section, although experiments were done over several months on three different instruments. However, correlations of CCS values were systematically higher than those for retention times, for example, the

median correlation for the HeLa runs between June 2018 and May 2019 is $r = 0.990$ for retention times and $r = 0.995$ for cross sections (based on 1264 peptides per pairwise comparison on average). Further, the upper triangle of the heatmap shows patches of similar color, unlike the mirrored positions in the lower triangle (Fig. 2a). This indicates chromatographic batch-effects resulting in non-linear shifts or changes in the peptide elution order. In contrast, the absence of similar patterns in the CCS comparisons supports our starting hypothesis that the ion mobility is largely independent of experimental circumstances.

Closer inspection of the variation in CCS values revealed mostly linear shifts, which do not affect the correlation coefficient. These shifts were only in the range from absolute 0 to 40 Å$^2$ (median 9.4 Å$^2$) even for very distant measurements, and they are mainly due to variations of the gas flow in the TIMS tunnel. Importantly, a linear alignment based on a few peptide CCS values almost completely corrects for these shifts (Methods, Fig. 2b). With such an alignment, CCS values can be compared across disparate datasets, which we did for all analyses shown here. Across the 347,885 peptide CCS values measured at least in duplicate, the median coefficient of variation (CV) was 0.4%, which highlights the excellent reproducibility of TIMS CCS measurements also over longer periods of time and across instruments (Fig. 2c). This may even be improvable as suggested by our previously reported CVs of 0.1% for replicate injections of a whole-proteome digest on a single instrument[10]. Reassuringly, we found an excellent correlation of $^{TIMS}CCS_{N2}$ values and drift

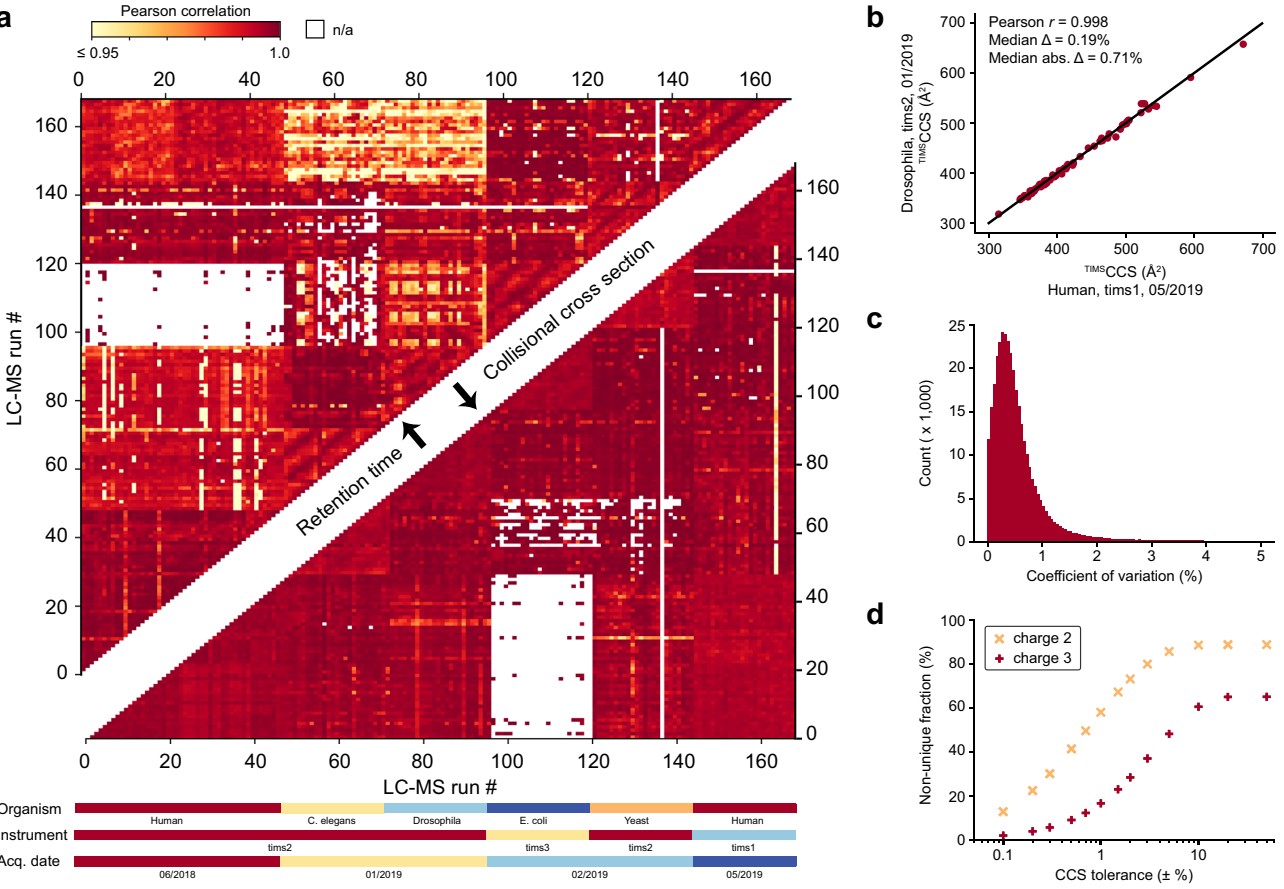

**Fig. 2 Precision, accuracy, and utility of experimental peptide CCS values. a** Color-coded pairwise Pearson correlation values of peptide retention time (upper triangular matrix) and CCS values (lower triangular matrix) between the 168 LC-MS/MS runs of fractionated tryptic digests. Experimental metadata are indicated below the x-axis. White (n/a) indicates less than 5 data points for pairwise comparison. **b** CCS values of shared tryptic peptides independently measured in two typical LC-MS runs of fractions from Drosophila and HeLa (n = 68). **c** CVs of repeatedly measured peptide CCS values in the full dataset (n = 347,885 peptides). **d** Specificity of combined peptide m/z and CCS information for doubly and triply charged peptides with C-terminal arginine or lysine (n = 324,246 and 112,015) with a fixed m/z tolerance of ±1.5 ppm and as a function of CCS tolerance. For details, see main text and Methods.

tube ion mobility experiments[42–44] (Pearson r = 0.997) with an average absolute deviation <2% (Supplementary Fig. 4).

To investigate the utility of the additional CCS information for peptide identification, we returned to Fig. 1e and defined tolerance windows in m/z and CCS dimensions for each peptide with C-terminal arginine or lysine as expected in tryptic digests (identified by MS/MS at an FDR < 1%). We then determined the fraction of windows in this map that were exclusively occupied by a single peptide, meaning a unique match between experimental measurement and our large peptide dataset (Fig. 2d). We set the mass tolerance at the median mass accuracy (±1.5 ppm) and varied the CCS tolerance separately for doubly and triply charged peptides, because they occupy different cross section areas (Methods). Without the CCS information, at ±50% tolerance, about 90% of the doubly charged and 67% of the triply charged peptides had at least one other peptide within 1.5 ppm distance ('non-unique'). The fraction of unique peptides increased once the CCS window was restricted to less than ±10%, in accordance with the roughly 20% spread of CCS values in Fig. 1e. Within three standard deviations (±1.5%) of the measured CCS values, about two-thirds of the doubly charged and 75% of the triply charged species were unique and these fractions increased progressively for narrower CCS windows. We thus conclude that ion mobility can substantially reduce the number of potential peptides that need to be considered, benefiting peptide

identification or MS1 level feature matching. At current CCS value accuracy, this is about a factor of two to three. As Fig. 2d also shows, an increase in accuracy down to 0.1% could make the large majority of peptides unique (56% for 2+ and 90% and 3+ in a ±0.5% CCS window).

**Dependence of CCS values on linear sequence determinants.** Having investigated the accuracy and utility of peptide CCS values, we asked whether a dataset of this scale could also shed a light on potential substructures in the m/z vs. ion mobility space and the relationships between linear peptide sequences and their corresponding gas-phase structures. In the m/z vs. CCS space of Fig. 1e, more compact conformations appear below and more extended confirmations appear above the overall trend lines for CCS values as a function of m/z.

We first explored whether amino acids with preferences for secondary protein structures[45], would also effect peptide ion structures in the gas phase and form clusters in this global view (Supplementary Fig. 5). This is a long-standing interest in ion mobility research and detailed studies of model peptides revealed that in particular helical structures can be stable in the gas phase[46–48]. Mapping the amino acids in each peptide sequence that favor helices in proteins, we found a tendency toward higher CCS with an increasing fraction of A, L, M, H, Q, and E. This suggests that such peptides, indeed, have a propensity to adopt

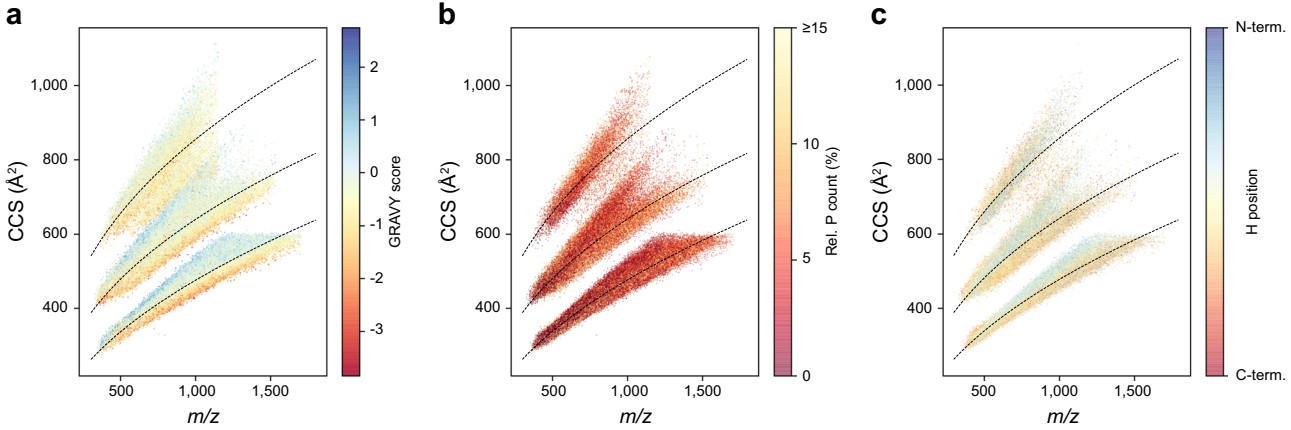

**Fig. 3 A global view on peptide cross sections. a** Mass-to-charge vs. collisional cross section distribution of all peptides in this study colored by the GRAVY hydrophobicity index ($n = 559{,}979$). **b** Subset of peptides with C-terminal arginine or lysine colored by the fraction of prolines in the linear sequence ($n = 452{,}592$). **c** Histidine-containing peptides of (**b**) colored by the relative position of histidine ($n = 171{,}429$). Trend lines (dashed) are fitted to the overall peptide distribution to visualize the correlation of ion mass and mobility in each charge state.

extended helical rather than more compact globular structures. In contrast, peptides with a high fraction of amino acids favoring turn structures (G, S, D, N, and P) tended to more compact conformations. Note, however, that these are subtle, population-wide effects. An interesting result was that peptides with <10% of the mostly non-polar amino acids V, I, F, T, and Y (favoring sheet structures in proteins) formed a narrow band of compact gas-phase conformations.

Such tendencies have been ascribed to intra-molecular interactions such as coulombic repulsion, charge solvation and hydrogen bonding[47–51]. We reasoned that the hydrophobicity of peptides could thus be a good indicator of these interactions in a global view. Indeed, the GRAVY score[52], a commonly used index of hydrophobicity, highlighted distinct areas of the $m/z$ vs. ion mobility space and within the CCS value distributions of each charge state, the peptides below the trend line had lower GRAVY scores than those above (Fig. 3a). The two major subgroups of the triply charged peptides also followed this trend in that hydrophobic peptides had a higher propensity to be in the upper population and vice versa. Interestingly, and perhaps counter-intuitively, this correlation was less apparent when comparing the relative bulkiness of amino acid residues even though these properties are related (Supplementary Fig. 6). These results are, however, in line with early work in ion mobility, indicating that non-polar amino acids contribute over-proportionately to the peptide CCS value[26,53] and stabilize helices in the absence of solvent[47]. When rotationally averaged, this results in larger, effective cross sections.

To resolve structural trends at the level of individual amino acids, we visualized their relative distribution in the same 2D space. Proline is unique due to its cyclic structure, which results in an inability to donate hydrogen bonds and to disruption of secondary structures in proteins. We found that peptides with more prolines had somewhat smaller CCS values on a global scale (Fig. 3b). In line with the above reasoning, this could be explained by a disruption of extended conformations and preference for globular ones.

A peptide's CCS value is not only determined by its amino acid composition, but also by its amino acid sequence. As a large-scale example of this, we generated complementary peptide sequences with lysine either at the N-terminus (LysN digestion) or at the C-terminus (LysC digestion). As described before[39], the two peptide populations are most distinct in triply charged species (Supplementary Fig. 7). Comparing 43,463 complementary sequences of

doubly charged peptides, we found changing CCS values in the range of −5% up to +10% with a slight median shift of about 1% toward higher CCS values for peptides with C-terminal lysine. The 14,388 triply charged species split in two sub-populations, with one maximum at about +1% similar to the doubly charged species and a second maximum at a shift of about +8%. This indicates that for the latter, switching the position of lysine from the C- to the N-terminus destabilizes the extended conformation. Assuming that the LysC peptides have a more extended conformation due to charge repulsion of the terminal charges, this again conforms to the above considerations.

We next investigated such effects in histidine-containing tryptic peptides, by color-coding them by their relative histidine position in the linear sequences (Fig. 3c). Peptides with histidines close to the N-terminus are more likely to adopt an extended conformation and peptides with histidines closer to the C-terminal lysine or arginine are more compact in the gas phase. This again emphasizes that the internal charge distribution and the ability to solvate charges intra-molecularly have a strong influence on peptide CCS values.

Although our analysis revealed interesting general trends and suggested common principles, it is challenging to combine them into robust models that rationalize the trends and determine the CCS value of a given peptide from its linear sequence. More importantly, peptide CCS values do not lend themselves to global ab initio calculations as this is beyond the capabilities of computational chemistry. To that end, we next turned to deep learning.

**Deep learning accurately predicts peptide CCS values**. To construct an accurate CCS predictor that can incorporate these large-scale peptide measurements, we decided to employ a flexible deep learning model. We set out to define a network architecture that is capable of learning a non-linear mapping function connecting the linear amino acid peptide sequence with associated charge states to the experimentally measured CCS value with the following properties: (i) Exploit the sequential structure of the data where each peptide is encoded as a string of amino acid sequences; (ii) Account for the influence of an amino acid in the context of the entire peptide sequence; and (iii) Process peptide sequences of arbitrary length. An architecture fulfilling those properties is a bi-directional LSTM network on top of the raw sequence followed by a two-layer multilayer perceptron (MLP)

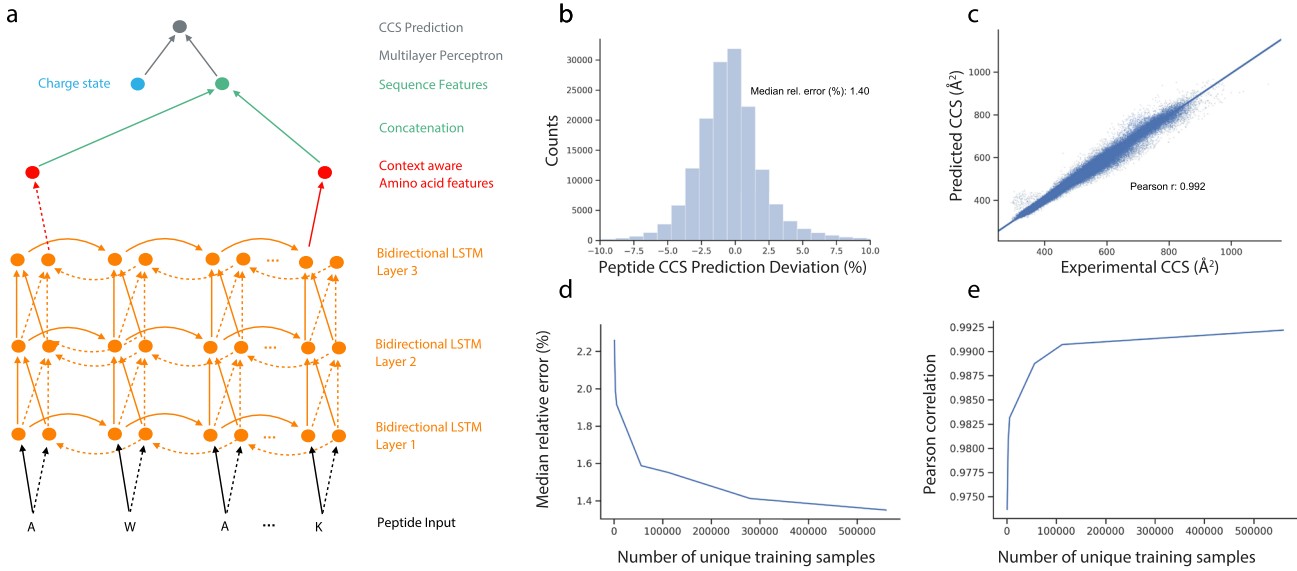

**Fig. 4 Deep learning peptide CCS values. a** Architecture of the neural network. Bi-directional long short-term model (LSTM): (i) amino acid sequence input, (ii) vectorization of amino acid information for processing, (iii) bi-directional LSTM layers, (iv) reduction to fixed length peptide feature vector by concatenating the last output neurons of both directional LSTMs, and (v) CCS prediction. **b** Relative deviation of predicted CCS values from an independent experimental validation dataset of synthetic peptides from the ProteomeTools project. **c** Correlation of predicted versus experimental CCS values ($n = 155,004$). **d** Dependence of the median relative error on training dataset size. **e** Same for Pearson correlation coefficient. Source Data are provided as a Source Data file.

(Fig. 4a, Methods). Similar models have already proven successful in proteomics[32,34,35]. The bi-directional LSTM layers enable the model to interpret each amino acid in the context of neighboring amino acids, while the following concatenation layer reduces the resulting N (sequence length) vectors into a single set of 256 features, together encoding the properties of the entirety of the peptide sequence. Together with the charge state, this vector constitutes the input to the MLP module for the final CCS value regression. The entire architecture is implemented with differentiable modules and is end-to-end trainable. We trained our model with the set of 559,979 unique CCS values from our experimental data of the five organisms.

Machine learning models, in particular deep learning models, can easily be over-fitted, resulting in poor generalization performance on new datasets. While holding out samples within the dataset helps, for a more rigorous safeguard, we acquired an independent additional dataset from the synthetic ProteomeTools peptides[54]. This yielded 155,004 unique peptide sequences as an external test set, which was never seen by the model during training. In this test set, our model reached a high accuracy with a 1.4% absolute median deviation and a Pearson correlation coefficient of 0.992 (Fig. 4b, c). For the subset of doubly charged peptides the median absolute deviation was 1.2%, and for triply and quadruply charged species it was 1.8% and 2.0%, respectively (Supplementary Fig. 8). Presumably as a result of an increasing number of accessible conformations, we found that the median absolute deviation increased from 1.2% for CCS values <400 Å$^2$, to 1.5% for CCS values between 400 and 800 Å$^2$ ($n = 129,710$) and 2.2% for 2580 peptides with CCS values >800 Å$^2$ (Supplementary Fig. 9). Of all predicted CCS values, 90% were within ±4.0% deviation from the experimental data. In comparison, the experimental median absolute deviation between tryptic peptides from ProteomeTools and endogenous peptides was 0.6% ($r = 0.995$, $n = 54,914$).

In our ProteomeTools data we also found a subset of 7% of the peptide sequences, for which MaxQuant identified at least one secondary feature with a CCS difference >2% relative to the most abundant feature. As we trained our model with CCS values of the latter, it is expected to predict the CCS value of the main conformation in such cases. However, for peptides with a more compact secondary conformation, we observed a bias toward lower CCS values and vice versa (Supplementary Fig. 10). Future prediction models may therefore benefit from considering multiple conformations, in particular for longer peptides and higher charge states.

To independently validate the accuracy of our predictions in a real-world example, we replaced experimental CCS values in a spectral library for DIA, built from the 24 HeLa fractions, with our predictions. We then used the experimental and the predicted libraries individually to re-analyze a triplicate diaPASEF experiment of a whole-proteome HeLa sample[55]. Targeted data analysis in the Spectronaut[56] software makes use of library values to score peptide signals and to restrict the data extraction window in the ion mobility dimension, thereby removing interfering signals from precursors with similar mass and retention time, but different ion mobility. The software automatically performs an alignment of the diaPASEF experiment to the library and optimized the median ion mobility extraction window to 0.07 and 0.09 Vs cm$^{-2}$ for the experimental and predicted library, respectively. The median absolute deviation of peptide ion mobility values were 0.74% and 0.93%. Overall, the experimental and predicted libraries performed very similarly, resulting in 7766 (experimental) and 7685 (predicted) identified protein groups on average (Supplementary Fig. 11).

Given that datasets in hundreds of thousands may still not be seen as large in deep learning, we next investigated the dependency between model accuracy in the test set and training dataset size (Fig. 4d, e). We observed a monotonous improvement in relative prediction accuracy as well as in the Pearson correlation with growing training dataset size. The model error decreased from 1.91% median relative error at 5600 samples to 1.42% for a set of 279,990 training samples, reflecting a substantial decrease in relative error of more than 20%. In contrast, moving from 279,990 samples to the full set of

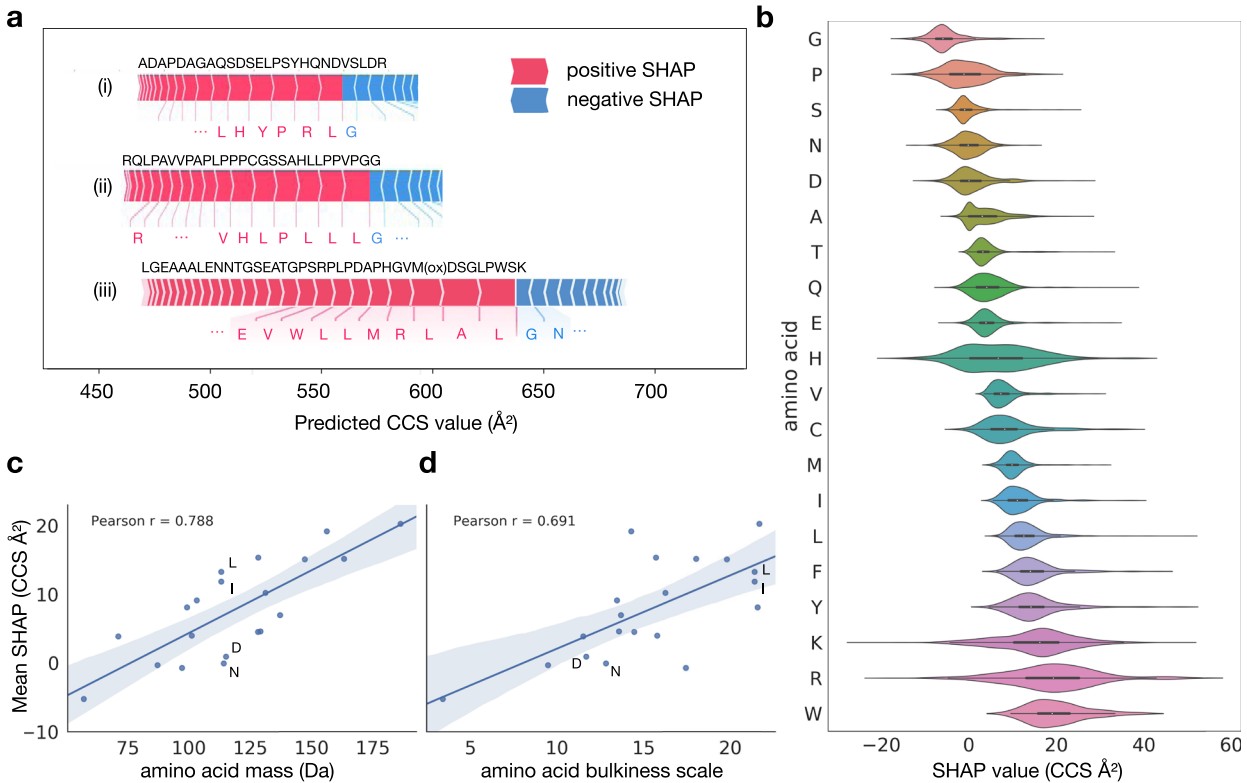

**Fig. 5 Explainable artificial intelligence reveals context-dependent amino acid contributions. a** Example peptide sequences with SHAP value attributions of the most influential amino acids in the linear sequence. **b** Amino acid-specific SHAP value distributions over the test dataset. Data are presented as violin plots showing kernel density estimates and boxplots with the following elements: median (center), 25th and 75th percentiles (lower and upper box limits), the 1.5× interquartile range (whiskers); $n = 100{,}000$ sequences. **c** Correlation between amino acid mass and mean SHAP value. **d** Correlation between amino acid bulkiness[59] and mean SHAP value.

559,979 samples resulted in a relative improvement of only 1.4% to a median relative error or 1.4%. These diminishing returns in accuracy of prediction indicated that the number of CCS values was sufficient—at least for currently achievable data quality.

**Resolving amino acids contributions.** Deep learning models are often deemed black boxes, as they are powerful predictors but learned relationships are typically hard to interpret. To make our model interpretable in relation to our experimental findings and to extract further molecular insights we calculated Shapley Additive Explanation (SHAP)[57,58] values for each amino acid in each sequence. In this case, SHAP values indicate the influence of a specific amino acid on the peptide CCS value by comparing it to reference values determined by randomly sampling sequences. This allowed us to interpret the CCS prediction for a peptide sequence by determining the individual, contextual attribution of each amino acid (Methods).

Figure 5a illustrates our analysis of sequence-specific amino acid SHAP values for three representative peptide sequences. In the regular tryptic peptide sequence (i), arginine and leucine with long side-chains shifted the prediction value to larger CCS as compared with a random sequence, while the smaller glycine contributed less than average. In the atypical peptide sequence (ii), the attribution of leucine was similar, however, the attribution of arginine was largely reduced in the N-terminal position. The context-dependent attribution of each amino acid was also evident from the long peptide sequence (iii), indicating a relatively large contribution of the small amino acid alanine to the prediction value. Interestingly, in this particular sequence, glutamic acid had a positive attribution, whereas asparagine

somewhat reduced the prediction value, despite the fact that both are similar in size and mass.

Plotting the aggregated SHAP value distribution over the entire test dataset for each individual amino acid, showed the expected relative order in terms of their average contribution (Fig. 5b): light and small amino acids such as glycine and proline had smaller SHAP values, whereas large and bulky amino acids such as tryptophan, arginine and lysine had larger attributions on average. In line with this observation, the average SHAP values correlated well with the amino acid mass and bulkiness[59], as indicated by Pearson correlation coefficients of 0.79 and 0.69, respectively (Fig. 5c, d). Deviations from these correlations, for example, for asparagine, aspartic acid, leucine, and isoleucine, which all have similar mass, could be explained by differences in their bulkiness and hydrophobicity, in line with our experimental results above. Collectively, these results highlight that our deep learning model learned plausible features, extracting related physical quantities on the level of individual amino acids automatically from the training data, even though we solely used the linear peptide sequence as an input.

Beyond the average values, the contribution of individual amino acids to a CCS prediction had vastly different values depending on their position in a sequence (Fig. 5b). Whereas the contributions of glycine, serine, glutamic acid, and methionine were quite constant, those of lysine, arginine, and histidine nearly varied over the entire range of observed SHAP values. In particular for histidine, this agrees with our empirical observation that the position in the linear sequence had a distinct effect on the cross section (Fig. 3c). We thus conclude that our model resolves substantial structural effects for some of the amino acids within each sequence to provide a very accurate CCS estimate for the entire peptide.

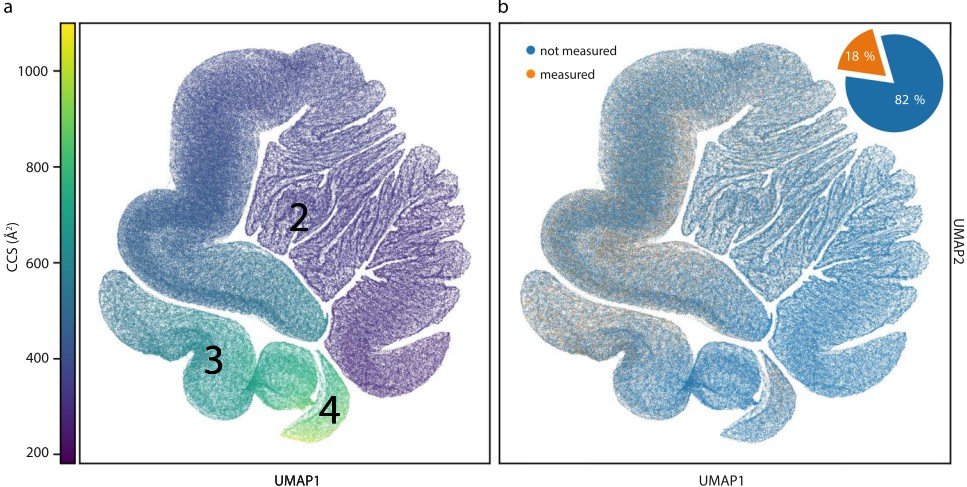

**Fig. 6 The human peptide CCS universe. a** Two-dimensional UMAP representation of 616,948 unique tryptic peptide sequences colored by their predicted CCS value. **b** Same UMAP plot. Peptide sequences with experimental values in this study are highlighted in orange (18%).

**Human whole-proteome level CCS prediction**. The human proteome gives rise to 616,948 unique tryptic peptide sequences (considering a minimum length of 7 amino acids and no missed cleavages), of which we measured about 18% in the course of this study. To investigate the entire peptide universe and to create a reference database of all tryptic peptides in the human organism, we next used our trained deep learning model to predict CCS values for the remaining 82%. Given the importance of charge in ion mobility and the fact that it does not follow from the linear sequence in a trivial manner, we first trained a second deep learning model on our experimental training data to also predict the charge state (Methods). We then fed each human peptide sequence together with its predicted charge state into the trained CCS model, resulting in a virtually complete compendium of human peptide CCS values (Supplementary Data 1).

To provide a bird's-eye view of the structure of these data, we visualized the data manifold learned in the last layer of the neural network, in which each sequence is described by a vector of 256 neural network features. These features represent all information relevant to the prediction and were used to regress the final CCS values. However, the data manifold is too high dimensional to be directly accessible to human interpretation, hence we used a non-linear dimension reduction algorithm (Uniform Manifold Approximation and Projection, UMAP[60]) to visualize the data in a 2D space. In this view, each point represents a single peptide sequence and each local structure represents classes of peptides with similar features. Distances in this space can be interpreted as similarities between sequences in terms of the features extracted by the network, meaning that sequences with similar gas-phase properties are close to each other. Figure 6a reveals that the neural network organized the data in three connected manifolds, in which the sequences are ordered in terms of their associated CCS value, starting with low CCS values (<300 Å$^2$) in the first cluster and increasing to high values (>900 Å$^2$) in the third cluster. Similar to the representation in $m/z$ vs. CCS space, we found that the main clusters were directly associated with the charge state and, within each charge state, there were apparent local structures.

Importantly, our experimental CCS values are distributed across the entire predicted peptide universe (orange and blue points in Fig. 6b), with very high densities in the CCS regions 400−800 Å$^2$, and lower densities in the region below 300 Å$^2$. This reassures that the depth of our experimental dataset was sufficient to sample the full feature space, and therefore suggests that our

model can be applied to predict CCS values of any tryptic peptide sequence with similar high accuracy.

## Discussion

Technological advances have rekindled the interest in IMS, which is now about to become mainstream in proteomic laboratories. Differential ion mobility spectrometers act as filters, only allowing selected ions to enter the mass spectrometer. In contrast, TIMS allows to measure ion mobility values and to derive CCS values that reflect an ion's size and shape. To investigate the benefit of this additional information in proteomics and making use of the speed and sensitivity of PASEF, we measured over two million CCS values of about 500,000 unique peptide sequences from five biological species. This covers a substantial proportion of the peptide space and is by far the most comprehensive dataset of CCS values to date.

This scale allowed us to first assess the analytical benefits of CCS values, which turn out to correspond to a roughly ten-fold increase in separation power. We further established that at an accuracy of 1%, the number of possible precursors of a peptide in a proteomics experiment decreases about two- to three-fold. Such an accuracy can be achieved with a simple linear re-calibration across distant measurements and different instruments. With this re-calibration, CCS values essentially become intrinsic properties of a molecule—meaning they do not depend on external circumstances—similar to their molecular weights, and unlike their retention times. In this regard, we note ongoing research on minimizing ion heating effects in TIMS measurements, as this may also influence the observed cross section or result in fragmentation before MS/MS, depending on instrument settings and space-charge effects[61–64]. However, results presented here and in other studies[15,22,65,66] indicate that $^{TIMS}$CCS values are generally in excellent agreement with the current gold-standard drift tube ion mobility.

The scale and uniformity of our dataset makes it a valuable resource to investigate fundamentals of peptide gas-phase structures in detail. Beyond the well-known correlation of CCS values with peptide mass, they also correlated with physicochemical amino acid properties such as hydrophobicity, while the contribution of certain amino acids varied based on their position in the sequence. While this scale allowed us to compare a multitude of different peptide sequences, a limitation of our analysis is that we considered only one CCS per peptide and charge state for

simplicity. However, ions from a single peptide may occur in multiple gas-phase conformations that can be resolved by IMS[50]. Even more information could thus be derived by resolving the ion-mobility fine structure, for example, of higher charge states[51] or proline-containing peptides[67]. As peptide CCS values in the gas phase are fully determined by their linear amino acid sequences, we reasoned that they should also be predictable with high accuracy. Indeed, after training our state-of-the-art deep learning model on our extensive dataset, it achieved a median accuracy of about 1% for independently measured synthetic peptides, close to the experimental uncertainty. Our model generalized very well to the extent that it accurately predicted CCS values even for unseen peptides, such as those from the 'missing genes' subset in ProteomeTools[54]. Adding even more data values would have diminishing returns, however, prediction accuracy could be further improved with even more consistent measurements and higher ion mobility resolution or by considering multiple conformations. To obtain a sufficient number of CCS values for deep learning, we trained and validated our model with complex samples of proteolytic digests and pooled synthetic peptides. In the future, this work could be complemented with manual investigation of isolated peptides, for example, to study mobility peak shapes and multiple conformations in more detail and independent of MS feature detection algorithms or other factors.

We also interrogated our deep learning model with regard to the determinants of its predictions with Shapley Additive Explanation (SHAP). Amino acids greatly differ in the extent to which their CCS contribution depends on their sequence context —ranging from almost none to a rather wide positive or negative contribution compared to an average amino acid. This highlights how our model, indeed, learned underlying principles. These could readily be extended to other peptide classes, such as modified[68] or cross-linked[69] peptides, using transfer learning[70], with little additional experimental effort.

Our study complements recent efforts in predicting properties of peptides on the basis of their sequences alone, especially those using deep learning for retention times and MS/MS spectra intensities[32,34,35]. Taken together, almost any peptide property relevant to proteomics workflows can now be predicted accurately, even in an ion mobility setup. Conceptually, this allows the community to nearly fully reconstruct the expected experimental values of a MS-based proteomics experiment, given a list of identified and quantified peptides. In more narrow terms, there is great potential to render time- and cost-intensive experimental libraries largely dispensable as exemplified here for diaPASEF. The CCS model presented here further extends the capabilities of such strategies to make full use of the ion mobility dimension. Similarly, predicted CCS values open up the possibility to reuse comprehensive community libraries such as the Pan Human library[71] for ion mobility-enhanced DIA or targeted workflows. We further envision that the combination of predicted CCS, retention time, and MS/MS spectra may improve scoring in database searches and narrow down the list of candidates. This is especially important in challenging applications such as peptidomics or proteomics of microbiomes[34] that have a very large search space. To foster its application and further developments, we make the source code available for training and predictions, in addition to the ready-to-use predictions of the human peptide universe included here.

## Methods

**Sample preparation**. The human HeLa cell line (S3, ATCC), *C. elegans* (N2 wild-type), *D. melanogaster* (CantonS), *E. coli* (XL1 Blue), and *Saccharomyces cerevisiae* (BY4741) were cultivated following standard protocols. All animal experiments

were performed in compliance with the institutional regulations of the Max Planck Institute of Biochemistry and the government agencies of Upper Bavaria. Whole organisms were first grinded in liquid nitrogen and cell pellets were directly suspended in lysis buffer with chloroacetamide (PreOmics, Germany) to simultaneously lyse cells, reduce protein disulfide bonds, and alkylate cysteine side chains[72]. The samples were boiled at 95 °C for 10 min and subsequently sonicated at maximum power (Bioruptor, Diagenode, Belgium). Proteolytic digestion was performed overnight at 37 °C by adding either (i) equal amounts of LysC and trypsin, (ii) LysC, or (iii) LysN in a 1:100 enzyme:protein (wt/wt) ratio. The resulting peptides were de-salted and purified via solid-phase extraction on styrenedivinylbenzene reversed-phase sulfonate (SDB-RPS) sorbent according to our 'in-StageTip' protocol (PreOmics). The dried eluates were reconstituted in water with 2% acetonitrile (ACN) and 0.1% trifluoroacetic acid (TFA) for further analysis. The synthetic ProteomeTools[54] peptides were reconstituted in the same buffer. To make the data comparable and reusable, we spiked iRT standards (Biognosys) into all samples.

**High-pH reversed-phase fractionation**. Peptide fractionation was performed at pH 10 on an EASY-nLC 1000 (Thermo Fisher Scientific, Germany) using a 30 cm × 250 μm $C_{18}$ reversed-phase column (PreOmics). Approximately 50 μg of peptides were separated at a flow rate of 2 μL min$^{-1}$ with a binary gradient starting from 3% B, which was linearly increased to 30% B within 45 min, to 60% B within 17 min, and to 95% B within 5 min before re-equilibration. Fractions were collected into 24 wells by switching the rotor valve of an automated concatenation system[73] (Spider fractionator, PreOmics) in 90 s intervals. Peptide fractions were vacuum-centrifuged to dryness and reconstituted in water with 2% ACN and 0.1% TFA.

**Liquid chromatography and mass spectrometry**. LC–MS was performed on an EASY-nLC 1200 (Thermo Fisher Scientific) system coupled online to a hybrid TIMS-quadrupole TOF mass spectrometer[10] (Bruker Daltonik timsTOF Pro, Germany) via a nano-electrospray ion source (Bruker Daltonik Captive Spray). Approximately 200 ng of peptides were separated on an in-house 45 cm × 75 μm reversed-phase column at a flow rate of 300 nL min$^{-1}$ in an oven compartment heated to 60 °C. The column was packed in-house with 1.9 μm $C_{18}$ beads (Dr. Maisch Reprosil-Pur AQ, Germany) up to the laser-pulled electrospray emitter tip. Mobile phases A and B were water and 80%/20% ACN/water (v/v), respectively, and both buffered with 0.1% formic acid (v/v). To analyze fractionated peptides from whole-proteome digests, we used a gradient starting with a linear increase from 5% B to 30% B over 95 min, followed by further linear increases to 60% B and finally to 95% B in 5 min each, which was held constant for 5 min before returning to 5% in 5 min and re-equilibration for 5 min. The pooled synthetic peptides were analyzed with a gradient starting from 5% B to 30% B in 35 min, followed by linear increases to 60% B and 95% in 2.5 min each before washing and re-equilibration for a total of 5 min.

The mass spectrometer was operated in data-dependent PASEF[13] mode with 1 survey TIMS-MS and 10 PASEF MS/MS scans per acquisition cycle. We analyzed an ion mobility range from $1/K_0 = 1.51$ to 0.6 Vs cm$^{-2}$ using equal ion accumulation and ramp time in the dual TIMS analyzer of 100 ms each. Suitable precursor ions for MS/MS analysis were isolated in a window of 2 Th for $m/z < 700$ and 3 Th for $m/z > 700$ by rapidly switching the quadrupole position in sync with the elution of precursors from the TIMS device. The collision energy was lowered stepwise as a function of increasing ion mobility, starting from 52 eV for 0–19% of the TIMS ramp time, 47 eV for 19–38%, 42 eV for 38–57%, 37 eV for 57–76%, and 32 eV until the end. We made use of the $m/z$ and ion mobility information to exclude singly charged precursor ions with a polygon filter mask and further used 'dynamic exclusion' to avoid re-sequencing of precursors that reached a 'target value' of 20,000 a.u. The ion mobility dimension was calibrated linearly using three ions from the Agilent ESI LC/MS tuning mix ($m/z$, $1/K_0$: 622.0289, 0.9848 Vs cm$^{-2}$; 922.0097, 1.1895 Vs cm$^{-2}$; and 1221.9906, 1.3820 Vs cm$^{-2}$). All experimental parameters with relevance to the measurement of CCS values are summarized in Supplementary Table 1.

**Data processing**. MS raw files were analyzed with MaxQuant[36,37] version 1.6.5.0, which extracts 4D isotope patterns ('features') and associated MS/MS spectra. The built-in search engine Andromeda[74] was used to match observed fragment ions to theoretical peptide fragment ion masses derived from in silico digests of a reference proteome and a list of 245 potential contaminants using the appropriate digestion rules for each proteolytic enzyme (trypsin, LysC or LysN). We allowed a maximum of two missing values and required a minimum sequence length of 7 amino acids while limiting the maximum peptide mass to 4600 Da. Carbamidomethylation of cysteine was defined as a fixed modification, and oxidation of methionine and acetylation of protein N-termini were included in the search as variable modifications. Reference proteomes for each organism including isoforms were accessed from UniProt (*Homo sapiens*: 91,618 entries, 2019/05; *E. coli*: 4403 entries, 2019/01; *C. elegans*: 28,403 entries, 2019/01; *S. cerevisiae*: 6049 entries, 2019/01; *D. melanogaster*: 23,304 entries, 2019/01). The synthetic peptide library (ProteomeTools[54]) was searched against the entire human reference proteome. The maximum mass tolerances were set to 20 and 40 ppm for precursor and fragment ions,

respectively. False discovery rates were controlled at 1% on both the peptide spectrum match and protein level with a target-decoy approach. The analyses were performed separately for each organism and each set of synthetic peptides ('proteotypic set', 'SRM atlas', and 'missing gene set'). To demonstrate the utility of CCS prediction, we re-analyzed three diaPASEF experiments from Meier et al.[55] with Spectronaut 14.7.201007.47784 (Biognosys AG), replacing experimental ion mobility values in the spectral library with our predictions. Singly charged peptide precursors were excluded from this analysis as the neural network was exclusively trained with multiply charged peptides.

**Bioinformatic analysis**. Bioinformatic analysis of the MaxQuant output files and data visualization was performed with Python version 3.6 employing the following packages: NumPy, pandas, SciPy[75], Biopython[76], Matplotlib, and Seaborn. Decoy database hits were excluded from the analysis as well as peptide features assigned with zero intensity values. Peptides can adopt multiple conformations, both in the liquid and in the gas phase. For simplification, we here selected only the most abundant feature for each modified peptide sequence and charge state per LC-TIMS-MS run. To account for experimental drifts in the measurements of $^{TIMS}$CCS values over time, we performed a hierarchical clustering (similar to[37]) and aligned all experiments by calculating pair-wise linear offsets ($y = x + b$) going from the closest to the most distant nodes. Multiple measurements of the same modified peptide and charge state in different LC-MS experiments were merged to one unique CCS value by calculating the mean. To perform nearest neighbor analysis in the $m/z$ vs. CCS space, we represented the data in a Kd-tree structure using the Chebyshev distance metric to define a rectangular area with a given mass and CCS tolerance surrounding a node of interest.

**Deep learning model for CCS prediction**. The deep learning model takes a raw (modified) peptide sequence as input. First, each amino acid gets one-hot encoded into a 26-dimensional vector representation for processing. This one-hot encoding also is applied to the elements '(ox)' and '(ac)', resulting in a total feature vector with dimension $L \times 26$ with $L$ being the length of a given peptide. This vector is connected to a two-layer bi-directional recurrent network with LSTM[77] units with 500 hidden nodes each, which extract context-based features for each individual amino acid. This feature embedding gets further reduced to a global 256-dimensional peptide feature vector by concatenating the last output neurons of both the LSTM networks aggregating from left or right over the sequence. This peptide feature vector is further concatenated with additional charge state of the sequence and then is fed to a logistic regression layer which regresses the expected CCS value for the sequence. The most significant hyperparameters, namely: number of hidden neurons, number of layers were chosen by running a small search in a first preliminary step on a validation set consisting of 10% of the training data. The combination of recurrent layers with the concatenation step allows the model architecture to process peptide sequences with arbitrary lengths. The final model is end-to-end optimized by an ADAM Optimizer on 559,979 unique CCS values (modified peptide sequence and charge state) and validated on 155,004 holdout peptides from the synthetic ProteomeTools library. The full framework is implemented using Python with TensorFlow[78] as the autograd library, enabling the neural network optimization. On an i7-4930K CPU machine equipped with an NVIDIA Geforce 1080 our model was trained within 8 h and the prediction of single peptide CCS values takes approximately 1 ms.

**Deep learning model for peptide charge state prediction**. To predict the most probable (most abundant) charge state from the linear peptide sequence, we built a charge prediction neural network which has the identical structure as our CCS prediction model. It takes the raw peptide sequence as input following the same one-hot encoding procedure and predicts a single associated charge value. We trained the charge prediction model on the same 559,979 unique training values and validated it on the holdout set of 155,004 peptides from ProteomeTools. The charge prediction model reaches a final accuracy of 93.5% for predicting the three observed charge states 2, 3, and 4.

**Analysis of amino acid feature attribution of the learnt network**. For a given sequence and its CCS prediction, every amino acid is associated with a SHAP value[57,58]. This SHAP value quantifies how the presence of the amino acid influences the final prediction. By the summation-to-delta property, the SHAP values are constrained in a way such that the sum of all SHAP values in a sequence results in the final CCS prediction. SHAP values are a unification of multiple existing approaches[79–83] for explaining predictions by feature attribution. For interpreting the predictions of our model we use the DeepExplainer from the official SHAP implementation (https://github.com/slundberg/shap). The DeepExplainer approximates SHAP values and is based on DeepLift[84]. Here the importance of individual features is approximated by comparing the model output for an input that contains the specific feature value to the model output where the feature is set to a reference value. A crucial step for this approach is to define the reference values. In our case, the inputs are sequences of one-hot-encoded amino acids and we use 128 randomly chosen background sequences from the dataset in order to define meaningful reference values for all neurons. In order to capture non-linearities, the DeepLift approach approximates feature attributions for every neuron in the model. It starts

at the output layer and propagates the values to the input by backpropagation, which is called applying the chain rule for multipliers in the original publication[81]. Applying this approach to the input sequences in our CCS model we are able to capture the SHAP value for an individual amino acid in a peptide sequence.

**Visualization of learnt network representation of the human proteome**. To visualize the 256-dimensional neural network feature space, we apply the UMAP[60] algorithm, which is a dimension reduction technique for general non-linear dimension reduction and it assumes uniform distribution of the data on a Riemannian manifold. Under certain conditions this manifold can be modeled with a fuzzy topological structure. The 2D embedding, which is used for visualization is found by searching for a low-dimensional projection of the data that has the closest possible equivalent fuzzy topological structure. Therefore, pairwise similarities between peptide sequences in the high-dimensional NN space approximately resemble positions in the low-dimensional embedding visualization.

**Reporting summary**. Further information on research design is available in the Nature Research Reporting Summary linked to this article.

## Data availability

The MS raw files and associated MaxQuant output files generated and analyzed throughout this study have been deposited at the ProteomeXchange Consortium via the PRIDE partner repository[85] with the dataset identifier PXD019086. The previously acquired HeLa data is available through the dataset identifier PXD010012. The diaPASEF raw files are available through the dataset identifier PXD017703. *H. sapiens* (taxon identifier: 9606), *S. cerevisiae* (taxon identifier: 559292), *D. melanogaster* (taxon identifier: 7227), *E. coli* (taxon identifier: 83333) and *C. elegans* (taxon identifier: 6239) proteome databases were downloaded from UniProt [https://www.uniprot.org]. Source data are provided with this paper.

## Code availability

The source code of our deep learning model and data analysis scripts are available on GitHub (https://github.com/theislab/DeepCollisionalCrossSection and https://github.com/mannlabs/DeepCollisionalCrossSection).

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

## Acknowledgements

We thank our colleagues in the department of Proteomics and Signal Transduction for help and discussions, and in particular I. Paron, B. Splettstößer, J. Müller, and A. Strasser for technical assistance. We acknowledge the ProteomeTools project led by B. Küster for providing a clone of the synthetic peptide library. This work was partially supported by the German Research Foundation (DFG-Gottfried Wilhelm Leibniz Prize granted to M.M., grant# MA 1764/2-1) and by the Max-Planck Society for the Advancement of Science. F.J.T. acknowledges support by the BMBF (grant #L031L0214A, grant# 01IS18036B and grant# 01IS18053A) and by the Helmholtz Association's Initiative and Networking Fund through Helmholtz AI (grant number: ZT-I-PF-5-01) and sparse2big (grant number ZT-I-007).

## Author contributions

F.M., A.B., and M.M. designed the proteomics experiments. F.M. and A.B. performed the experiments. F.M., A.B., and M.M. analyzed the data and interpreted the results. E.V. and M.T.S. contributed to the data analysis. N.D.K., with contributions from F.J.T., designed and developed the deep learning model as well as the prediction interpretation and visualization pipeline. J.M.W. performed neural network training runs and supported N.D.K. in integrating the feature attribution functionality. F.M, N.D.K., F.J.T., and M.M. wrote the manuscript. F.J.T. and M.M. supervised the project.

## Funding

## Competing interests

F.J.T. reports receiving consulting fees from Roche Diagnostics GmbH and Cellarity Inc., and ownership interest in Cellarity, Inc. and Dermagnostix. The other authors declare no competing interests.
