## [Peer Review File · Nature Communications]

REVIEWER COMMENTS

Reviewer #1: Remarks to the Author:

This manuscript presents a valuable research and resource for the community. An analysis of collisional cross section (CCS) results from whole-proteome digests of five organisms with trapped ion mobility spectrometry (TIMS) and parallel accumulation serial fragmentation was performed. The authors used the experimental results to implement and train a deep recurrent neural network that predicts CCS values from the peptide sequence.

The deep learning model with input and network architecture is well explained, including the software framework used for the implementation. Multiple statistical analysis, validations and visualizations are well presented and described.

Recommendations are included in the following paragraphs to further improve the manuscript.

The authors have addressed the 2 comments made by the previous reviewer.

For comment #1 regarding the generalizability of the data presented in the manuscript compared to other ion mobility platforms, the authors have performed a correlation of their TIMS CCS results against 24 drift tube CCS results in common with two publicly available studies and added the Supplementary Figure 4. However, it is likely that a more comprehensive comparison could have been performed using the experimental CCS database compiled from those studies and others in <https://mcleanresearchgroup.shinyapps.io/CCS-Compendium/> (J. A. Picache, B. S. Rose, A. Balinski, K. L. Leaptrot, S. D. Sherrod, J. C. May and J. A. McLean, Chem. Sci., 2019, 10, 983–993).

For comment #2 about how prediction of peptide CCS may benefit proteomics experiments, the authors have added results and discussions about re-using existing large-scale community LC-MS/MS libraries acquired without ion mobility separation that can now be used for ion-mobility proteomics experiments by predicting the CCS using the authors' tool and incorporating CCS into the library. In addition, they replaced experimental CCS values in a data-independent acquisition spectral library with their predictions and show that both experimental and predicted libraries performed very similar in the number of identified protein groups in a HeLa digest and added the Supplementary Figure 7.

I believe it would be useful to add a few sentences clarifying the identification procedure in this comparison, stating that the 2 libraries with experimental and predicted CCS values were used individually to perform identification on a LC-IMS-MS/MS run based on targeted data extraction (which is different from the untargeted identification performed initially for the training dataset with the MaxQuant software), as well as a comment on how the identification software (Spectronaut) used the CCS in the library to score the peptide identifications (e.g. as a filter with a tolerance?) and how the CCS accuracy could impact the identification results.

The source code of the deep learning model is not available. The availability is cited as <https://github.com/theislab>, however, this is a general link and I could not find any package related to deep learning and CCS. Please add the full link to the specific repository. Furthermore, there is no mention about the computing resources and time that was required to train the model.

Since the study specifically used 360 LC-MS/MS runs and two million CCS values, the phrase “from a million training samples” in the title could be misleading from the analytical and experimental perspective. It is technically correct from the machine learning or statistical perspective but using, for example, “training measurements” or “training values” instead of “training samples” could be more appropriate for the MS community.

The validation against the “ProteomeTools Library” is not well described.

First, I suggest avoiding the term “library” here. I was initially confused to whether this was existing data or data from a sample analyzed by the authors. For example, instead of using

“we acquired an independent additional dataset from the synthetic ProteomeTools library” use “we acquired an independent additional dataset from the synthetic ProteomeTools peptides”

Second, I could not see the data in PRIDE due to errors unzipping the files. I suggest adding a supplementary csv file where the predicted and experimental CCS values are included for each peptide.

Third, I recommend adding a sentence about the limitations of training and validating based on tryptic digests of complex samples or pooled synthetic peptides compared to individual analysis of synthetic peptides and pure CCS calculations. I believe that a future validation by analyzing individual peptides and calculating CCS values independently of other factors such as possible bias introduced by feature finding or MS/MS matching algorithms will be beneficial for the acceptance and adoption of the CCS prediction tool by the community.

Finally and related to the previous point, another limitation that should be mentioned is the arbitrary selection of the most abundant feature per charge state in the cases where peptides occur in more than one conformation and have multiple peaks, since this is neglecting the power of ion mobility to reveal multiple conformers that may exist, and for example, may be relevant to a disease state or mechanism of the system under study.

Reviewer #2 (Remarks to the Author):

Meier et al. measured more than two million CCS values of peptides and ~500,000 unique peptides from five different biological samples. Utilizing these experimental CCS values, they developed a deep-learning based CCS prediction approach to predict CCS values. The CCS prediction method demonstrated excellent performances for external validation sets. The authors also systemically investigated the determinants for CCS values of peptides, such as secondary structures, hydrophobicity, amino acid composition, sequence, et al. Overall, this work provided an important resource for proteomics and IM-MS community. Before the publication, I wish the following comments could be addressed.

1. The authors observed 0-40 Å² variations for measured CCS values in different measurements. These shifts were corrected using the linear alignment approach in this work. To make this data set universal for the IM-MS community, I think the alignment procedures needs to be standardized and easy to reproduce. The demo data and examples should be provided. Then, common users can compare their own data with the CCS values (both experimental and predicted ones) reported in this work.

2. As described in the manuscript, ~570,000 unique CCS values were screened and retained from more than 2.5 million CCS values. I guess that many of them were repeatedly measured. How did the authors merge these values into one unique value or filter unwanted ones? These need to be described in details in Methods part.

3. The CCS data for model training and test in this work should be provided in supplementary for data reproducibility. Or a publicly accessible web server is fine too.

4. Supplementary Figure 1 showed that about 10% peptides occurred with multiple CCS values presumably from different conformations. I understood that the authors chose the most abundant peptide for the model training. For these peptides with multiple conformations, can author compare the predicted CCS values from deep learning model? It is important to investigate these peptides, otherwise it will introduce false negatives using the CCS match for peptide identification or in the targeted proteomics workflow shown in Figure S7.

5. I observed that the training set had different compositions in peptide sequences (79% C-terminal, 21% N-terminal) and charge states (69% double charge, 25% triple charge, 4% quadruple charge). Therefore, does the specific composition affect the model training and the applicability of prediction? Or are the used composition in peptide sequences universal for most biological organism?

6. I noticed that most peptides in the training data had the experimental CCS values between 400-800 Å² (Figure 6b). Can authors evaluate the prediction performance for different CCS intervals (e.g., <400 Å², 400-800 Å², >800 Å²)?

7. For the application, the authors demonstrated an interesting example in Figure 2d. In the experimental sets, the fraction of unique peptides increased with the application of a CCS match tolerance. I wonder whether it is also effective if the MS/MS match was used in addition to CCS match?

8. I would welcome some discussions about the potential of predicted CCS values for peptide identification in proteomics workflow, especially combining the CCS match with predicted retention times and predicted MS/MS spectra.

Zheng-jiang Zhu

Reviewer #3 (Remarks to the Author):

The manuscript “Deep learning the collisional cross sections of the peptide universe from a million training samples” by Meier et al discusses the use of collisional cross sections for proteomics studies.

The manuscript largely separates into two parts. In the first part, the authors argue on the basis of their extensive data that cross section values are likely reflective of the amino acid sequence of peptides. In the second part, they train a neural network using their large data set that reproduces/predicts peptide cross sections. These findings are significant, because based on the data shown in the manuscript it appears highly likely that cross section information could aide the sequence identification in proteomics studies. I therefore recommend acceptance subject to a revision.

I have two major points that I would like the authors to address prior to acceptance

1) I wonder how strongly ion heating in the TIMS influences their cross sections and the conclusions made. Here, I am less concerned with the accuracy of the cross section calibration, instead I refer to the possibility of peptide ions to pick up vibrational energy inside or outside the TIMS and subsequently to dissociate or rearrange. This would potentially skew their results and lead to an incorrect neural network model. Morsa and DePauw et al recently (*Analytical Chemistry*, 2020) discussed ion heating in TIMS and proposed ion temperatures of >500 K. Such high temperatures can easily induce fragmentation of peptide bonds, in particular for the quadruply charged ions and/or the proline-containing species. I recommend that the authors discuss this possibility of ion heating, or, conversely discuss why ion heating might not take place in the current work.

2) It is not typical that a peptide exists only in a single conformation in ion mobility spectrometry; this holds in particular for multiply charged peptides (see, for example, the body of work on bradykinin by Clemmer). Hence, I would expect that a significant portion of the data points shown in Figures 1-3 were obtained from ion mobility spectra with more than just 1 peak per amino acid sequence. The authors do not discuss the presence of multiple cross sections in their manuscript; it is unclear how their neural network would cope with the situation of multiple cross sections for a single input amino acid sequence; and it is unclear if the disregard of multiple cross sections would bias their neural network; nor is it clear how this challenge would potentially introduce errors into the interpretation of proteomics studies that build upon the neural network developed in this manuscript.

Minor points:

Page 1, Line 27: "ions' rotationally averaged gas phase conformation" This wording is inaccurate. The observed cross section is rotationally averaged (external rotations) even for a single conformation or a rigid structure. When the ion undergoes molecular dynamics during the ion mobility measurement, then the cross section is additionally averaged over the sampled conformations.

Page 1, Line 30. "have quite distinct shapes by this feature". This wording is inaccurate or unclear. Do the authors mean to say that different types of molecules have different ion mobilities/cross sections? Or do the authors mean to say that different types of molecules show different trends in their mobilities depending on the compound mass (see the papers by McLean on the various trendlines for peptides, lipids, sugars, etc.)?

Figure 3a: the GRAVY score essentially is a hydrophobicity scale. Is it really true that the cross sections of peptides increase with their hydrophobicity? Or could this be just as well be explained by the bulkiness of the side chains?

Point-by-point response to the reviewer comments on “Deep learning the collisional cross sections of the peptide universe from a million training samples”

We highly appreciate the reviewer’s encouraging evaluation of our manuscript and their very constructive feedback, which helped us to substantially strengthen our manuscript. We carefully addressed all comments as detailed point-by-point below. In the course of the revision, we have re-trained our deep learning model, added further analyses and clarified the text following the expert suggestions. As remarked by all three reviewers, the fact that peptides can occur in multiple gas phase conformations poses a challenge to any CCS prediction model and proteomics in general. We have now investigated this in more detail and discuss our results in the revised manuscript. With these improvements, we hope that the reviewers and the editor find our revised manuscript suitable for publication in *Nature Communications*.

Reviewer #1 (Remarks to the Author, see also attached file):

This manuscript presents a valuable research and resource for the community. An analysis of collisional cross section (CCS) results from whole-proteome digests of five organisms with trapped ion mobility spectrometry (TIMS) and parallel accumulation serial fragmentation was performed. The authors used the experimental results to implement and train a deep recurrent neural network that predicts CCS values from the peptide sequence.

The deep learning model with input and network architecture is well explained, including the software framework used for the implementation. Multiple statistical analysis, validations and visualizations are well presented and described.

Recommendations are included in the following paragraphs to further improve the manuscript.

We thank the reviewer for their kind evaluation of our work and highlighting its value for the scientific community. The excellent recommendations helped us to improve our manuscript as detailed below.

The authors have addressed the 2 comments made by the previous reviewer.

For comment #1 regarding the generalizability of the data presented in the manuscript compared to other ion mobility platforms, the authors have performed a correlation of their TIMS CCS results against 24 drift tube CCS results in common with two publicly available studies and added the Supplementary Figure 4. However, it is likely that a more comprehensive comparison could have been performed using the experimental CCS database compiled from those studies and others in <https://mcleanresearchgroup.shinyapps.io/CCS-Compendium/> (J. A. Picache, B. S. Rose, A. Balinski, K. L. Leaprot, S. D. Sherrod, J. C. May and J. A. McLean, *Chem. Sci.*, 2019, 10, 983–993).

We agree with the reviewer that our analysis could have been strengthened by comparing a larger number of peptide CCS values across different ion mobility devices. Unfortunately,

publicly available reference ion-nitrogen CCS values of electrosprayed peptides are very limited. The McLean lab's unified CCS compendium provides a high-quality resource for CCS values of over 3,000 ion species from a large variety of biochemical compounds (csv database, downloaded 2020/11/06) and we added this reference to our revised manuscript. Filtering for compounds with the keyword "Tryptic Peptide" resulted in 138 entries. Only 8 of these were protonated species with charge state 2+ and they were all from the Stow et al. data, which was already included in our comparison.

For comment #2 about how prediction of peptide CCS may benefit proteomics experiments, the authors have added results and discussions about re-using existing large-scale community LC-MS/MS libraries acquired without ion mobility separation that can now be used for ion-mobility proteomics experiments by predicting the CCS using the authors' tool and incorporating CCS into the library. In addition, they replaced experimental CCS values in a data-independent acquisition spectral library with their predictions and show that both experimental and predicted libraries performed very similar in the number of identified protein groups in a HeLa digest and added the Supplementary Figure 7.

I believe it would be useful to add a few sentences clarifying the identification procedure in this comparison, stating that the 2 libraries with experimental and predicted CCS values were used individually to perform identification on a LC-IMS-MS/MS run based on targeted data extraction (which is different from the untargeted identification performed initially for the training dataset with the MaxQuant software), as well as a comment on how the identification software (Spectronaut) used the CCS in the library to score the peptide identifications (e.g. as a filter with a tolerance?) and how the CCS accuracy could impact the identification results.

We agree and thank you for these suggestions. We revised our text accordingly:

"To independently validate the accuracy of our predictions in a real-world example, we replaced experimental CCS values with our predictions in a spectral library for data-independent acquisition (DIA). We then used the experimental and the predicted libraries individually to analyze a triplicate diaPASEF experiment of a whole-proteome HeLa sample⁵⁴. Targeted data analysis in the Spectronaut software makes use of library values to score peptide signals and to restrict the data extraction window in the ion mobility dimension, thereby removing interfering signals from precursors with similar mass and retention time, but different ion mobility. The software automatically performs an alignment of the diaPASEF experiment to the library and optimizes the median ion mobility extraction window to 0.07 Vs cm⁻² and 0.09 Vs cm⁻² for the experimental and predicted library, respectively. The median absolute deviation of peptide ion mobility values were 0.74% and 0.93%. Overall, the experimental and predicted libraries performed very similarly, resulting in 7,766 (experimental) and 7,685 (predicted) identified protein groups on average (Supplementary Fig. 11)."

The source code of the deep learning model is not available. The availability is cited as <https://github.com/theislab>, however, this is a general link and I could not find any package related to deep learning and CCS. Please add the full link to the specific repository.

Furthermore, there is no mention about the computing resources and time that was required to train the model.

Thank you for pointing this out. We have now made the source code public and added the full link to the data availability section (<https://github.com/theislab/DeepCollisionalCrossSection>). The repository contains all information to either predict CCS values using our pre-trained model or to train a new model with other data.

We also added a statement on computing resources and time to the Methods section as suggested:

“On an i7-4930K CPU machine equipped with an NVIDIA Geforce 1080 our model was trained within 8 hours and the prediction of single peptide CCS values takes approximately 1 ms.”

Since the study specifically used 360 LC-MS/MS runs and two million CCS values, the phrase “from a million training samples” in the title could be misleading from the analytical and experimental perspective. It is technically correct from the machine learning or statistical perspective but using, for example, “training measurements” or “training values” instead of “training samples” could be more appropriate for the MS community.

We agree with the reviewer that ‘samples’ could cause confusion and we thus changed the title to ‘...from a million experimental values’.

The validation against the “ProteomeTools Library” is not well described.

First, I suggest avoiding the term “library” here. I was initially confused to whether this was existing data or data from a sample analyzed by the authors. For example, instead of using “we acquired an independent additional dataset from the synthetic ProteomeTools library” use “we acquired an independent additional dataset from the synthetic ProteomeTools peptides”

We agree that the suggested wording is more precise and have changed “library” to “peptides”. Thank you.

Second, I could not see the data in PRIDE due to errors unzipping the files. I suggest adding a supplementary csv file where the predicted and experimental CCS values are included for each peptide.

We apologize for this inconvenience and have now re-uploaded the entire data set. We confirmed that all result files related to the Proteome Tools data are downloadable and the zip files can be extracted. We tested “Results_ProteomeTools_Proteotypic.zip” on Windows 7 with WinRAR 5.71 and Windows 10 using the built-in tool as well as 7-Zip 19.00.

As suggested by the reviewer, we added a Supplementary csv file to our revised manuscript that contains the source data for Figure 4.

Third, I recommend adding a sentence about the limitations of training and validating based on tryptic digests of complex samples or pooled synthetic peptides compared to individual analysis of synthetic peptides and pure CCS calculations. I believe that a future validation by analyzing individual peptides and calculating CCS values independently of other factors such as possible bias introduced by feature finding or MS/MS matching algorithms will be beneficial for the acceptance and adoption of the CCS prediction tool by the community.

We agree with the reviewer that detailed studies of individual peptides are invaluable to gain further insight into gas phase structures of peptides and may be used to validate specific observations in large data sets such as ours. We discuss this point in our revised manuscript:

“To obtain a sufficient number of CCS values for deep learning, we trained and validated our model with complex samples of proteolytic digests and pooled synthetic peptides. In the future, this work could be complemented with manual investigation of isolated peptides, for example to study mobility peak shapes and multiple conformations in more detail and independent of MS feature detection algorithms or other factors.”

Finally and related to the previous point, another limitation that should be mentioned is the arbitrary selection of the most abundant feature per charge state in the cases where peptides occur in more than one conformation and have multiple peaks, since this is neglecting the power of ion mobility to reveal multiple conformers that may exist, and for example, may be relevant to a disease state or mechanism of the system under study.

We agree with the reviewer that this is a limitation of our current approach and added the following to the main text.

“A limitation of our analysis is that we considered only one CCS per peptide and charge state for simplicity. However, ions from a single peptide may occur in multiple gas phase conformations that can be resolved by ion mobility spectrometry. Even more information could thus be derived from resolving the ion mobility fine structure, for example of higher charge states or proline-containing peptides.”

Please see also our response to point 4 of reviewer #2 and point 2 of reviewer #3.

Reviewer #2 (Remarks to the Author):

Meier et al. measured more than two million CCS values of peptides and ~500,000 unique peptides from five different biological samples. Utilizing these experimental CCS values, they developed a deep-learning based CCS prediction approach to predict CCS values. The CCS prediction method demonstrated excellent performances for external validation sets. The authors also systemically investigated the determinants for CCS values of peptides, such as

secondary structures, hydrophobicity, amino acid composition, sequence, et al. Overall, this work provided an important resource for proteomics and IM-MS community. Before the publication, I wish the following comments could be addressed.

We highly appreciate Dr. Zhu's profound expertise and have carefully addressed all points below.

1. The authors observed 0-40 Å² variations for measured CCS values in different measurements. These shifts were corrected using the linear alignment approach in this work. To make this data set universal for the IM-MS community, I think the alignment procedures needs to be standardized and easy to reproduce. The demo data and examples should be provided. Then, common users can compare their own data with the CCS values (both experimental and predicted ones) reported in this work.

Thank you. We have uploaded Jupyter Notebooks to <https://github.com/MannLabs/> to facilitate reproduction of our analysis and further development.

2. As described in the manuscript, ~570,000 unique CCS values were screened and retained from more than 2.5 million CCS values. I guess that many of them were repeatedly measured. How did the authors merge these values into one unique value or filter unwanted ones? These need to be described in details in Methods part.

Thank you for pointing this out. We specified this in the Methods section of our revised manuscript:

"Multiple measurements of the same modified peptide and charge state in different LC-MS experiments were merged to one unique CCS value by calculating the mean."

3. The CCS data for model training and test in this work should be provided in supplementary for data reproducibility. Or a publicly accessible web server is fine too.

We fully agreed and deposited the data as source data and to the GitHub repository together with the source code (<https://github.com/theislab/DeepCollisionalCrossSection> and <https://github.com/mannlabs/DeepCollisionalCrossSection>).

4. Supplementary Figure 1 showed that about 10% peptides occurred with multiple CCS values presumably from different conformations. I understood that the authors chose the most abundant peptide for the model training. For these peptides with multiple conformations, can author compare the predicted CCS values from deep learning model? It is important to investigate these peptides, otherwise it will introduce false negatives using the CCS match for peptide identification or in the targeted proteomics workflow shown in Figure S7.

Thank you for suggesting this very interesting analysis, which we have now included in our revised manuscript.

Please note that in Supplementary Fig. 1 we have considered only peptides that were identified by their fragmentation pattern with high confidence, which may underestimate the overall presence of multiple conformations. This is because, even though PASEF acquires MS/MS spectra at >100 Hz, this speed is still not sufficient to fragment all precursors in a complex tryptic digest and for low-abundance features, the signal quality might be insufficient for identification. This effect should be reduced in less complex samples and we have thus turned to our data set of the synthetic ProteomeTools peptides. For 24% of all identified modified sequence- and charge-unique peptides, MaxQuant assigned more than one feature in at least one LC-MS experiment, which were either multiple peaks in retention time or ion mobility dimension. Interestingly, 7% had at least one secondary feature with a CCS difference >2% relative to the most abundant feature, indicating that TIMS can resolve multiple gas phase conformations in an LC-MS setup. Regardless of secondary conformations, our model is expected to predict the CCS value of the most abundant feature as it is exclusively trained on these. However, investigating the prediction error for this subset of peptide sequences, we found larger deviations as compared with the overall distribution (median absolute error 1.23% vs. 1.18% for charge 2 and 2.82% vs. 1.82% for charge 3). Interestingly, for peptides with a more compact secondary conformation, our predictions were shifted to lower CCS values and vice versa (new Supplementary Fig. 10). This analysis shows the challenge to predict CCS value for such peptides accurately, but also suggests a great potential to improve predictions by considering multiple conformations. We believe that the CCS resource included here provides an excellent basis for the community to follow up on this in future work. The revised manuscript has been changed accordingly.

Supplementary Fig. 10: CCS value prediction accuracy by charge state for peptide sequences detected with multiple features in LC-TIMS-MS experiments of synthetic ProteomeTools peptides. x is the relative distance of the most distant secondary feature to the most abundant feature in the CCS dimension within one LC-TIMS-MS experiment. 78 values are outside the shown x-axis range.

5. I observed that the training set had different compositions in peptide sequences (79% C-terminal, 21% N-terminal) and charge states (69% double charge, 25% triple charge, 4% quadruple charge). Therefore, does the specific composition affect the model training and the applicability of prediction? Or are the used composition in peptide sequences universal for most biological organism?

This is an excellent observation. Indeed, the predictions could be less accurate for sub-classes with fewer data points due to the fact that the model has been trained with less samples for these. Importantly, this will not affect the performance for other classes. The observed imbalance is in part due to the design of our experiment as we prepared most proteolytic digests with trypsin and LysC, resulting in the majority of peptides having C-terminal arginine and lysine residues. The observed charge state distribution is very typical for such samples and dependent on the peptide sequence and length. In that regard, the distribution of tryptic peptides generated from five different organisms is representative for the vast majority of proteomics experiments. To test whether there is an effect of charge state on the performance of our model, we investigated the prediction accuracy for the synthetic ProteomeTools peptides separately for charge states 2, 3 and 4. These results are now included in the main text and the new Supplementary Fig 8. Please note that the ProteomeTools set does not include LysN peptides (N-terminal lysine and unspecified C-term), which is why we did not perform the same analysis for this subset.

“In this test set, our model reached a high accuracy with a 1.4% absolute median deviation and a Pearson correlation coefficient of 0.992 (Fig 4b,c). For the subset of doubly charged peptides the median absolute deviation was 1.2%, and for triply and quadruply charged species it was 1.8% and 2.0%, respectively (Supplementary Fig. 8). Potentially indicating an increasing structural variability, we found that the median absolute accuracy increased from 1.2% for CCS values $<400\text{\AA}^2$, to 1.5% for CCS values between $400\text{-}800\text{\AA}^2$ and 2.2% for CCS value $>800\text{\AA}^2$ (Supplementary Fig. 9). Of all predicted CCS values, 90% were within $\pm 4.0\%$ deviation from the experimental data. In comparison, the experimental absolute median deviation between tryptic peptides from ProteomeTools and endogenous peptides was 0.6% ($r = 0.995$, $n = 54,914$).”

Supplementary Fig. 8. Relative deviation of predicted CCS values from an experimental validation dataset of synthetic peptides from the ProteomeTools project by charge state.

6. I noticed that most peptides in the training data had the experimental CCS values between $400\text{-}800\text{\AA}^2$ (Figure 6b). Can authors evaluate the prediction performance for different CCS intervals (e.g., $<400\text{\AA}^2$, $400\text{-}800\text{\AA}^2$, $>800\text{\AA}^2$)?

Thank you, we have extended our performance evaluation also in response to your point directly above and added this analysis to our revised manuscript:

“In this test set, our model reached a high accuracy with a 1.4% absolute median deviation and a Pearson correlation coefficient of 0.992 (Fig 4b,c). For the subset of doubly charged peptides the median absolute deviation was 1.2%, and for triply and quadruply charged species it was 1.8% and 2.0%, respectively (Supplementary Fig. 8). Presumably as a result of an increasing number of accessible conformations, we found that the median absolute deviation increased from 1.2% for CCS values $<400\text{\AA}^2$, to 1.5% for CCS values between $400\text{-}800\text{\AA}^2$ ($n=129,710$) and 2.2% for 2580 peptides with CCS values $>800\text{\AA}^2$ (Supplementary Fig. 9). Of all predicted CCS values, 90% were within $\pm 4.0\%$ deviation from the experimental data. In comparison, the experimental absolute median deviation between tryptic peptides from ProteomeTools and endogenous peptides was 0.6% ($r = 0.995$, $n = 54,914$).”

Supplementary Fig. 9. Relative deviation of predicted CCS values from an experimental validation dataset of synthetic peptides from the ProteomeTools project by CCS value.

7. For the application, the authors demonstrated an interesting example in Figure 2d. In the experimental sets, the fraction of unique peptides increased with the application of a CCS match tolerance. I wonder whether it is also effective if the MS/MS match was used in addition to CCS match?

Thank you for highlighting this analysis. In this example, we only considered peptides that were already identified by their MS/MS spectrum at a false discovery rate $<1\%$ to have a ground-truth for matches. We added this information to the main text:

“To investigate the utility of the additional CCS information for peptide identification, we returned to Fig. 1e and defined tolerance windows in m/z and CCS dimensions for each peptide with C-terminal arginine or lysine as expected in tryptic digests (identified by MS/MS at an $FDR < 1\%$).”

Please see also our response to your point below regarding the potential of CCS values for peptide identification in proteomics.

8. I would welcome some discussions about the potential of predicted CCS values for peptide identification in proteomics workflow, especially combining the CCS match with predicted retention times and predicted MS/MS spectra.

This is a very interesting point and we elaborate more on this in the discussion of our revised manuscript as suggested:

“Our study complements recent efforts in predicting properties of peptides on the basis of their sequences alone, especially those using deep learning for retention times and MS/MS spectra intensities^{32,34,35}. Taken together, almost any peptide property relevant to proteomics workflows can now be predicted accurately, even in an ion mobility set-up. Conceptually, this allows the community to nearly fully reconstruct the expected experimental values of a MS-based proteomics experiment, given a list of identified and quantified peptides. [...] We further envision that the combination of predicted CCS, retention time and MS/MS spectra may improve scoring in database searches and narrow down the list of candidates. This is especially important in challenging applications such as peptidomics or proteomics of microbiomes³⁴ that have a very large search space. To foster applications of our CCS model and further developments, we make the source code available for training and predictions, in addition to the ready-to-use predictions of the human peptide universe included here.”

Zheng-jiang Zhu

Reviewer #3 (Remarks to the Author):

The manuscript “Deep learning the collisional cross sections of the peptide universe from a million training samples” by Meier et al discusses the use of collisional cross sections for proteomics studies.

The manuscript largely separates into two parts. In the first part, the authors argue on the basis of their extensive data that cross section values are likely reflective of the amino acid sequence of peptides. In the second part, they train a neural network using their large data set that reproduces/predicts peptide cross sections. These findings are significant, because based on the data shown in the manuscript it appears highly likely that cross section information could aide the sequence identification in proteomics studies. I therefore recommend acceptance subject to a revision.

We thank the reviewer for the in-depth study of our manuscript and the excellent and constructive feedback provided. Addressing the points detailed below clearly strengthened our analysis and interpretation of the ion mobility data in the revised manuscript.

I have two major points that I would like the authors to address prior to acceptance

1) I wonder how strongly ion heating in the TIMS influences their cross sections and the conclusions made. Here, I am less concerned with the accuracy of the cross section calibration, instead I refer to the possibility of peptide ions to pick up vibrational energy inside or outside

the TIMS and subsequently to dissociate or rearrange. This would potentially skew their results and lead to an incorrect neural network model. Morsa and DePauw et al recently (Analytical Chemistry, 2020) discussed ion heating in TIMS and proposed ion temperatures of >500 K. Such high temperatures can easily induce fragmentation of peptide bonds, in particular for the quadruply charged ions and/or the proline-containing species. I recommend that the authors discuss this possibility of ion heating, or, conversely discuss why ion heating might not take place in the current work.

Thank you for raising this important point. We are very aware of the current discussion in the ion mobility community about ion heating in TIMS and the recent contributions to Analytical Chemistry (Morsa *et al.* 2020 and Bleiholder *et al.* 2020) as well as JASMS (Naylor *et al.* 2020). The consensus of these studies is that, while TIMS is inherently a “soft” ion mobility technique that can successfully retain native structures, ion heating may occur depending on experimental conditions. Bleiholder *et al.* argue that strong ion heating is not intrinsic to TIMS but results from space-charge effects and RF-heating. In line with this, Yu *et al.* (Mol. Cell. Proteomics 2020) reported an increased number of gas phase fragmentation events before MS/MS (semi-tryptic peptides and water-losses) for longer TIMS accumulation times up to 400 ms in proteomics experiments. We chose our experimental parameters to achieve optimal proteomics performance (100 ms accumulation time) and thus the reviewer is correct that there is a possibility for ion heating. We added this discussion and the references to our revised manuscript as suggested:

“In this regard, we note ongoing research on minimizing ion heating effects in TIMS measurements, as this also influence the observed cross section, depending on instrument settings and space-charge effects^{59-62.}”

Please note that we included only peptides in our analysis that fully match the expected protein sequence and cleavage pattern of the corresponding protease in each experiment (e.g. full tryptic peptides). Further, we required a precursor mass accuracy <20 ppm and controlled false discovery rates <1% at the peptide spectrum match and protein levels. Any peptide fragmentation or rearrangement upstream the collision cell is therefore unlikely to result in a valid spectrum match and lead to a wrong sequence-to-CCS assignment.

2)It is not typical that a peptide exists only in a single conformation in ion mobility spectrometry; this holds in particular for multiply charged peptides (see, for example, the body of work on bradykinin by Clemmer). Hence, I would expect that a significant portion of the data points shown in Figures 1-3 were obtained from ion mobility spectra with more than just 1 peak per amino acid sequence. The authors do not discuss the presence of multiple cross sections in their manuscript; it is unclear how their neural network would cope with the situation of multiple cross sections for a single input amino acid sequence; and it is unclear if the disregard of multiple cross sections would bias their neural network; nor is it clear how this challenge would potentially introduce errors into the interpretation of proteomics studies that build upon the neural network developed in this manuscript.

We fully agree that peptides can occur in multiple gas phase conformations, and we estimate that at least ten percent of the peptides in our data set have multiple distinct conformations (please see also Supplementary Figure 1 and our response to Reviewer #2, point 4). However, a conclusive analysis at this scale remains challenging given limitations in the sensitivity of the instrument, ion mobility resolution, feature detection algorithms (reporting only one distinct value for one feature) and the fact that we included only confidently identified peptide features. Nevertheless, we are convinced that our publicly available data set provides a highly valuable resource for the community to investigate this phenomenon in more detail.

For simplicity and as a first step, we restricted our analysis to the most abundant feature per modified peptide sequence and charge state in a single LC-TIMS-MS experiment (page 3, lines 29-31). This is in line with previous work in this area and we note that currently all proteomics approaches assume a single CCS value for each peptide, for example in spectral libraries for data-independent acquisition. Computationally, there is also a clear advantage of assigning one discrete value to one sequence. In our case, the deep learning model is expected to predict the CCS value of the most abundant gas phase conformation, regardless of how many other conformations may exist. Our new results included in Supplementary Fig. 9 support this, even though we observed a lower prediction accuracy for peptides with multiple conformers. This could be a hint that this peptide property is encoded in the linear sequence and could potentially be predicted or, vice versa, predicting multiple conformations could increase the accuracy of the predictions. These are compelling directions of future research.

We have clarified this in our revised manuscript, also taking into account the additional comments of Reviewers #1 and #2 on this point.

“In our ProteomeTools data we also found a subset of 7% of the peptide sequences, for which MaxQuant identified at least one secondary feature with a CCS difference >2% relative to the most abundant feature. As we trained our model with CCS values of the latter, it is expected to predict the CCS value of the main conformation in such cases. However, for peptides with a more compact secondary conformation, we observed a bias towards lower CCS values and vice versa (Supplementary Fig. 10). Future prediction models may therefore benefit from considering multiple conformations, in particular for longer peptides and higher charge states.”

We do not expect that this introduces a particular experimental bias in proteomics experiments. This is based on the assumption that the relative abundance of the different gas phase conformations remains stable, at least to the extent that experimental parameters are not changed in a way that alters the energy distribution (related to the point just above). The high reproducibility of CCS values demonstrated here across many experiments and instruments suggests that this is a valid assumption.

Minor points:

Page 1, Line 27: “ions’ rotationally averaged gas phase conformation” This wording is inaccurate. The observed cross section is rotationally averaged (external rotations) even for a single conformation or a rigid structure. When the ion undergoes molecular dynamics during the ion mobility measurement, then the cross section is additionally averaged over the sampled conformations.

This is correct, thank you. We re-phrased this sentence in our revised manuscript:

“...which is the rotational average of an ions’ gas phase conformation^{21,22}.”

Page 1, Line 30. “have quite distinct shapes by this feature”. This wording is inaccurate or unclear. Do the authors mean to say that different types of molecules have different ion mobilities/cross sections? Or do the authors mean to say that different types of molecules show different trends in their mobilities depending on the compound mass (see the papers by McLean on the various trendlines for peptides, lipids, sugars, etc.)?

Thank you, we re-phrased this sentence referencing the work by McLean and colleagues:

“The CCS intrinsically depends on the ion structure, which is also illustrated by the fact that different classes of biomolecules (e.g. metabolites, carbohydrates, peptides) show different trends in their ion mobilities as a function of molecular mass²³.”

Figure 3a: the GRAVY score essentially is a hydrophobicity scale. Is it really true that the cross sections of peptides increase with their hydrophobicity? Or could this be just as well be explained by the bulkiness of the side chains?

This is a very interesting point that arises from our observation in the data. In fact, bulkiness and hydrophobicity are linked in that bulky side chains tend to be hydrophobic and both properties likely influence the gas phase structure. Of note, hydrophobicity and bulkiness factors in the literature are derived from experiments in the liquid or solid phase, and may be different in the gas phase. As such, our analysis can only be an approximation. To address this point, we have added a new analysis of our data in which we split peptides into quantiles according to their relative deviation from the CCS vs. m/z trend line of doubly and triply charged species. As expected, both GRAVY score and a bulkiness score (Zimmermann et al., 1968) were higher for peptides above the trend lines (larger CCS) as compared with peptides below the trend lines (smaller CCS). However, this effect was stronger for hydrophobicity, in particular for triply charged peptides. This observation is in line with early work by Clemmer and others, which also attributes larger CCS values to more hydrophobic peptides. We added this analysis in our revised manuscript:

“We reasoned that the hydrophobicity of peptides could thus be a good indicator of these interactions in a global view. Indeed, the GRAVY score⁵¹, a commonly used index of hydrophobicity, highlighted distinct areas of the m/z vs ion mobility space and within the CCS value distributions of each charge state, the peptides below the trend line had lower GRAVY scores than those above (Fig. 3a). The two major subgroups of the triply-charge peptides also

followed this trend in that hydrophobic peptides had a higher propensity to be in the upper population and vice versa. Interestingly, and perhaps counter-intuitively, this correlation was less apparent when comparing the relative bulkiness of amino acid residues even though these properties are related (Supplementary Figure 6). These results are, however, in line with early work in ion mobility, indicating that non-polar amino acids contribute over-proportionately to the peptide CCS value^{26,52} and stabilize helices in the absence of solvent⁴⁶. When rotationally averaged, this results in larger, effective cross sections.”

Supplementary Fig. 6. Global correlations of peptide CCS values and physicochemical properties. **a**, Mass-to-charge vs. collisional cross section distribution of doubly charged peptides color-coded in quantiles by the relative deviation from the trend line. **b**, Violin plots of the GRAVY scores in each quantile. **c**, Violin plots of the average amino acid bulkiness in each quantile. **d-f**, Same as **a-c** but for triply charged peptides.

REVIEWERS' COMMENTS

Reviewer #1 (Remarks to the Author):

The authors have properly addressed all reviewer's concerns and have taken into account the suggestions.

Reviewer #2 (Remarks to the Author):

The authors have addressed all my comments. I would be happy to see the manuscript published now.

Point-by-point response to the reviewer comments on “Deep learning the collisional cross sections of the peptide universe from a million experimental values”

Reviewer #1 (Remarks to the Author):

The authors have properly addressed all reviewer's concerns and have taken into account the suggestions.

Reviewer #2 (Remarks to the Author):

The authors have addressed all my comments. I would be happy to see the manuscript published now.

We wish to thank all reviewers once more for their critical insight and support of our manuscript.